# ITPNet: Towards Instantaneous Trajectory Prediction for Autonomous Driving

## Abstract

Trajectory prediction of moving traffic agents is crucial for the safety of autonomous vehicles, whereas previous approaches usually rely on sufficiently long-tracked locations (e.g., 2 seconds) to predict the future locations of the agents. However, in many real-world scenarios, it is not realistic to collect adequate observations for moving agents, leading to the collapse of most prediction models. For instance, when a moving car suddenly appears and is very close to an autonomous vehicle because of the obstruction, it is quite necessary for the autonomous vehicle to quickly and accurately predict the trajectories of the car with limited tracked trajectories. In light of this, we focus on investigating the task of instantaneous trajectory prediction, i.e., two tracked locations are available during inference. To this end, we put forward a general and plug-and-play instantaneous trajectory prediction approach, called ITPNet. At its heart, we propose a backward forecasting mechanism to reversely predict the latent feature representations of unobserved historical trajectories of the agent based on its two observed locations and then leverage them as complementary information for future trajectory prediction. Moreover, due to the inevitable existence of noise and redundancy in the predicted latent feature representations, we further devise a Noise Redundancy Reduction Former (NRRFormer) module, which attempts to filter out noise and redundancy from a sequence of unobserved trajectories and integrate the filtered features and the observed features into a compact query representation for future trajectory predictions. In essence, ITPNet can be naturally compatible with existing trajectory prediction models, enabling them to gracefully handle the case of instantaneous trajectory prediction. Extensive experiments on the Argoverse and nuScenes datasets demonstrate ITPNet outperforms the baselines by a large margin and shows its efficacy with different trajectory prediction models.

## 1 Introduction

Predicting the future trajectories of dynamic traffic agents is a critical task for autonomous driving, which can be beneficial to the downstream planning module of autonomous vehicles. In recent years, many trajectory prediction methods have been proposed in computer vision and machine learning communities (Wang et al., 2023; Park et al., 2023; Zhou et al., 2023; Zhu et al., 2023; Chen et al., 2021; Gu et al., 2022; Xu et al., 2022; Wang et al., 2022a; Meng et al., 2022). Among these methods, they usually need to collect sufficiently long tracked trajectories (typically, 2 to 3 seconds) of an agent, in order to accurately predict its future trajectories. Recent advances have shown promising performance in trajectory prediction by learning from these adequate observations.

However, when facing real-world self-driving scenarios, it is often difficult to have sufficient observations for accurate trajectory prediction. For instance, due to the obstruction, a moving car might suddenly appear and be very close to the autonomous vehicle. At this moment, the autonomous vehicle does not have enough time to collect adequate tracked trajectories of the car to accurately predict the car's future trajectories. Such a case will cause the collapse of the aforementioned prediction models due to the lack of information. To verify this point, we perform a typical trajectory prediction method, HiVT (Zhou et al., 2022), with different settings on the Argoverse dataset (Chang et al., 2019). As shown in the left part of Figure 1(a), if we use 20 tracked locations as the inputs of the prediction model during both training and test phases as in (Zhou et al., 2022), the prediction results are 0.698 and 1.053 in terms of minADE@6 and minFDE@6, respectively. However, if we

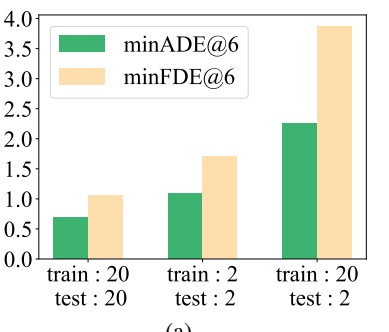 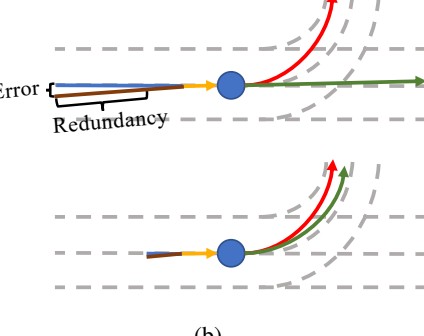

(a)  (b)

Figure 1: (a) Results of HiVT (Zhou et al., 2022) in terms of minADE@6 and minFDE@6 on the validation set of Argoverse (Chang et al., 2019) with different tracked locations as inputs during training and testing. The value in the horizontal axis denotes the number of tracked locations. (b) Future predictions (shown in green) when utilizing different lengths of predicted unobserved trajectories. The observed trajectories are shown in orange, the predicted unobserved trajectories are shown in brown, the ground-truth unobserved trajectories are shown in blue, and the ground-truth future trajectories are shown in red.

set only 2 tracked locations as the inputs of the model during testing, the model will degrade sharply, no matter if the number of tracked locations is 2 or 20 during the training phase. Thus, it is essential to study the trajectory prediction task, when tracked trajectories are very limited.

In light of this, we focus on studying the task of instantaneously predicting future trajectories of moving agents, under the assumption of only 2 trajectory points available. Recently, Sun et al. (2022a) proposes a trajectory prediction method based on momentary observations. However, this method mainly focuses on the trajectory prediction of pedestrians, which has not been explored for other moving agents. In addition, the input of their model is the RGB image which usually contains abundant context and semantic information. Thus, it is much easier for the model to predict future trajectories using RGB images, compared to only several discrete trajectory points. Moreover, Monti et al. (2022) design a knowledge distillation mechanism for trajectory prediction based on limited observed locations and achieves promising results. Since the method needs to pre-train a teacher model, and learns a student model distilling knowledge from the teacher model, which largely increases the computational complexities.

To this end, we propose a general and principled approach, called ITPNet, for instantaneous trajectory prediction by only two observed trajectory locations. The key to the success of ITPNet is to train a predictor to backwardly predict the latent feature representations of unobserved historical trajectories of the agent based on its two observed trajectories. The additional information contained in the predicted unobserved trajectory features assists observed trajectory features in better predicting future trajectories. Nevertheless, we find that as we increase the number of backwardly predicted unobserved trajectory locations, the model's performance initially improves but subsequently deteriorates, as illustrated in Table 3. We analyze two primary factors that impede the utilization of more unobserved trajectory features: One is the noise brought by inaccurate prediction of the unobserved trajectory features. The other is a negative impact on the trajectory prediction due to the intrinsic redundant information. Let's consider a scenario where a vehicle travels straightly for a while and then suddenly executes a turn. In such a case, a longer historical trajectory may erroneously boost the model's confidence in the vehicle continuing straight in the future, as depicted in the upper portion of Figure 1(b). Conversely, a shorter unobserved historical trajectory with less redundancy tends to yield more accurate predictions because it maintains lower confidence in the vehicle's persistence in a straight trajectory and, instead, maintains higher confidence in the vehicle's persistence in a turning trajectory, as shown in the lower portion of Figure 1(b). Thus, how to remove noisy and redundant information from the predicted features of the unobserved trajectories becomes the key to success in instantaneous trajectory prediction.

In view of this, we devise a Noise Redundancy Reduction Former (NRRFormer) module and integrate it into our framework. NRRFormer can filter out noise and redundancy from a sequence of predicted unobserved latent features, and effectively fuse the filtered unobserved latent features

with the observed features by a compact query embedding to acquire the most useful information for future trajectory prediction. It is worth noting that our ITPNet is actually plug-and-play, and is compatible with existing trajectory prediction models, making them the kinds that can gracefully deal with the instantaneous trajectory prediction problem.

Our main contributions are summarized as following: 1) We propose a backward forecasting mechanism to reconstruct unobserved historical trajectory information for instantaneous trajectory prediction, mitigating the issue of lack of information due to only two observed locations. 2) We devise a Noise Redundancy Reduction Former (NRRFormer), which can remove noise and redundancy among the predicted unobserved feature representations to further improve the prediction performance. 3) We perform extensive experiments on two widely used benchmark datasets, and demonstrate our ITPNet can outperform the baselines in a large margin. Moreover, we show the efficacy of ITPNet, combined with different trajectory prediction models.

## 2 RELATED WORKS

### 2.1 TRAJECTORY PREDICTION WITH SUFFICIENT OBSERVATION

In recent years, many trajectory prediction approaches have been proposed (Girgis et al., 2021; Gilles et al., 2022; Makansi et al., 2021; Casas et al., 2020a; Sun et al., 2022b; Cheng et al., 2023; Bae et al., 2023; Bae & Jeon, 2023; Choi et al., 2023). In the early stage of trajectory prediction, studies such as (Alahi et al., 2016; Gupta et al., 2018) usually rely solely on observation points and adopt simple social pooling methods to capture interactions between agents. To capture the map information, including occupancy or semantic information, (Bansal et al., 2018; Phan-Minh et al., 2020; Mohamed et al., 2020) propose to use Convolutional Neural Networks (CNN) to encode map images. In addition, (Gao et al., 2020; Liang et al., 2020) incorporate the information of lanes and traffic lights on the map in the form of vectors. Recently, numerous methods have been proposed to fully exploit the interaction information between nearby agents, including implicit modeling by graph neural networks (Casas et al., 2020a; Li et al., 2019; Salzmann et al., 2020; Casas et al., 2020b) and attention mechanisms (Nayakanti et al., 2022; Liu et al., 2021; Ngiam et al., 2022; Li et al., 2020), and explicit modeling (Sun et al., 2022b). To handle the uncertainty of road agents, researchers propose to generate multi-modal trajectories using various approaches, including GAN-based methods (Kosaraju et al., 2019; Sadeghian et al., 2019; Gupta et al., 2018), VAE-based methods (Lee et al., 2017; 2022), flow-based methods (Zhang et al., 2022; Liang et al., 2022), and diffusion models (Gu et al., 2022; Mao et al., 2023; Jiang et al., 2023). Among them, one typical approach is to establish a mapping between future trajectories and latent variables, producing multiple plausible trajectories by sampling the latent variable. In addition, goal-based methods have become popular recently (Zhao et al., 2021; Gu et al., 2021; Zeng et al., 2021; Wang et al., 2022b; Mangalam et al., 2021; Aydemir et al., 2023), which first generates multi-modal goals by sampling (Zhao et al., 2021) or learning (Wang et al., 2022b), and then predict future trajectories conditioned on the goals.

Although these methods have shown promising performance in trajectory prediction, they usually learn depending on sufficiently long-tracked trajectories. As aforementioned, these methods degrade severely or even collapse when the number of tracked trajectories is limited. Different from these works, we attempt to address the task of instantaneously predicting the future trajectories of moving agents, under the condition that only two trajectory locations are observable.

### 2.2 TRAJECTORY PREDICTION WITH INSTANTANEOUS OBSERVATION

Predicting the future trajectories of a moving agent by its limited tracked trajectory points remains a challenging problem. Recently, Sun et al. (2022a) proposes an approach to integrate the velocity of agents, social and scene contexts information, and designs a momentary observation feature Extractor (MOE) for pedestrian trajectory prediction. The input of MOE contains image frames from videos which usually contain abundant semantic information. Thus, it is much easier to predict future trajectories using image frames than that using several discrete trajectory points. Moreover, since this method is mainly designed for predicting trajectories for pedestrians, what is the performance on other moving agents, e.g., cars, is worth to be further verified. Monti et al. (2022) proposes a knowledge distillation approach using few observations as input, with the goal of lowering the influence of noise introduced by the machine perception side (i.e., incorrect detection and tracking).

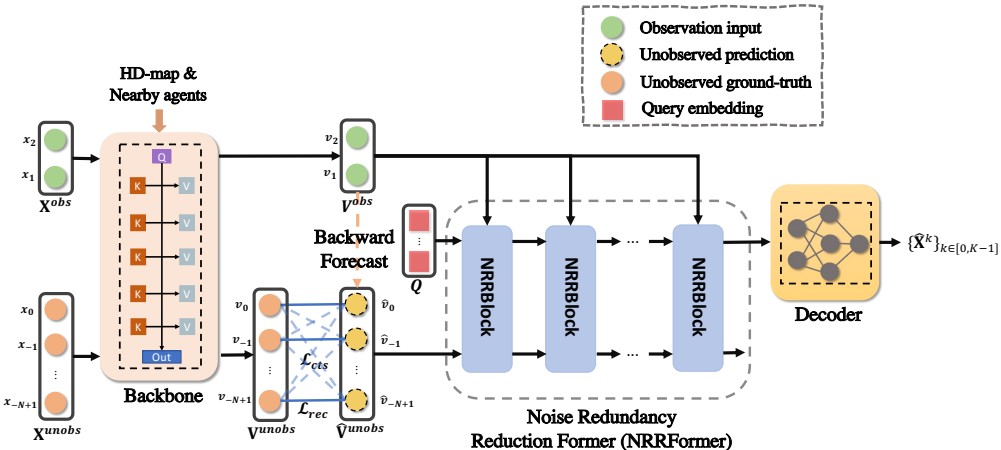

Figure 2: Overview of our ITPNet framework. ITPNet mainly consists of two modules: 1) We propose a backward forecasting mechanism that attempts to reconstruct the latent feature representations $\mathbf{V}^{unobs}$ of previous unobserved trajectories $\mathbf{X}^{unobs}$ by the two observed trajectories locations $\mathbf{X}^{obs}$. 2) We devise a Noise Redundancy Reduction Former to filter out noise and redundancy in the predicted latent feature representations $\hat{\mathbf{V}}^{unobs}$, and both the resulting filtered features and the observation features $\mathbf{V}^{obs}$ are integrated into a compact query embedding $\mathbf{Q}$. Finally, the query embedding is sent to the decoder to instantaneously predict future trajectories $\{\hat{\mathbf{X}}^k\}$.

As we know, knowledge distillation-based approaches generally need to pre-train a teacher model, and then distill knowledge from the teacher model to help the student model learn, which makes this kind of method computationally expensive.

## 3 PROPOSED METHOD

### 3.1 PROBLEM DEFINITION

We denote a sequence of observed states for a target vehicle as $\mathbf{X}^{obs} = \{x_1, x_2\}$, where $x_i \in \mathbb{R}^2$ is the $i$-th location of the agent. Moreover, we also denote the sequence of previous unobserved locations of the agent as $\mathbf{X}^{unobs} = \{x_{-N+1}, x_{-N+2}, \cdots, x_0\}$, where $N$ is the total number of unobserved locations. The ground-truth future trajectories are denoted as $\mathbf{X}^{gt} = \{x_3, x_4, ..., x_{2+M}\}$, where $M$ is the length of ground-truth future trajectory. Our goal is to predict future possible $K$ trajectories $\{\hat{\mathbf{X}}^k\}_{k \in [0, K-1]} = \{(\hat{x}_3^k, \hat{x}_4^k, ..., \hat{x}_{2+M}^k)\}_{k \in [0, K-1]}$, as in multi-model trajectory prediction methods (Gupta et al., 2018; Kosaraju et al., 2019; Lee et al., 2022). Differently, we attempt to leverage merely two observed locations $\mathbf{X}^{obs}$ for instantaneous trajectory prediction during inference, in contrast to previous methods utilizing sufficient tracked locations (typically, 20 observed locations on the Argoverse dataset (Chang et al., 2019)).

### 3.2 OVERALL FRAMEWORK

Figure 2 illustrates an overview of our proposed framework. We first feed the observed trajectories $\mathbf{X}^{obs}$ into a backbone (e.g., HiVT (Zhou et al., 2022)) to obtain the latent feature representations $\mathbf{V}^{obs} = \{v_1, v_2\}$. Based on this representation $\mathbf{V}^{obs}$, we then attempt to backwardly predict the latent feature representations $\hat{\mathbf{V}}^{unobs} = \{\hat{v}_{-N+1}, \hat{v}_{-N+2}, ..., \hat{v}_0\}$ of unobserved historical trajectories $\mathbf{X}^{unobs}$. Considering that the predicted unobserved feature representations $\hat{\mathbf{V}}^{unobs}$ inevitably contain redundant and noisy information as mentioned above, we design a Noise Redundancy Reduction Former (NRRFormer) module to filter out this information from a predicted feature sequence. Subsequently, the filtered features are combined with the observed features to generate a compact query embedding $\mathbf{Q}$. The query embedding $\mathbf{Q}$ is then sent to the decoder for future trajectory predictions. Since the backbone in our framework is arbitrary, our method is plug-and-play, and is compatible with existing trajectory prediction models, enabling them to gracefully adapt to the scenario of only

two observed locations. Next, we mainly introduce the backward forecasting and the NRRFormer in detail.

### 3.3 BACKWARD FORECASTING

When given only two tracked trajectories $\mathbf{X}^{obs}$, one major issue we face is the lack of information, making existing trajectory prediction approaches degraded sharply. To alleviate this problem, we propose to backwardly predict the latent feature representations of previous unobserved trajectories, and then leverage them as additional information for future trajectory prediction.

First, we can obtain the latent feature representations $\mathbf{V}^{obs}$ of the tracked trajectories $\mathbf{X}^{obs}$ via a backbone $\Phi$:

$$\mathbf{V}^{obs} = \{v_1, v_2\} = \Phi(\mathbf{X}^{obs}; \phi), \tag{1}$$

where $v_i \in \mathbb{R}^d$ is the latent feature representation of the $i$-th location of the agent, and $d$ is the dimension of the feature. The backbone $\Phi$ is parameterized by $\phi$, and can be an arbitrary trajectory prediction model, e.g., HiVT (Zhou et al., 2022) and LaneGCN (Liang et al., 2020) used in this paper. It is worth noting that our method is plug-and-play, since any trajectory prediction model can serve as the backbone.

After that, we attempt to backwardly predict the latent feature representations $\widehat{\mathbf{V}}^{unobs}$ on the basis of $\mathbf{V}^{obs}$, addressing the issue of the lack of information. To this end, we introduce two self-supervised tasks: the first one is the reconstruction of the latent feature representations, and the loss function is designed as:

$$\mathcal{L}_{rec} = \mathcal{J}(\mathbf{V}^{unobs}; \widehat{\mathbf{V}}^{unobs}), \tag{2}$$

where $\mathbf{V}^{unobs} = \Phi(\mathbf{X}^{unobs}; \phi)$ is the ground-truth latent feature representations of previous unobserved trajectories, and can be taken as a self-supervised signal, and $\mathcal{J}$ is a function to measure the distance between $\mathbf{V}^{unobs}$ and $\widehat{\mathbf{V}}^{unobs}$. $\widehat{\mathbf{V}}^{unobs}$ are the predicted features, obtained by:

$$\widehat{\mathbf{V}}^{unobs} = \Psi(\mathbf{V}^{obs}; \psi), \tag{3}$$

where $\Psi$ is a network parameterized by $\psi$. In this paper, we make use of a LSTM (Hochreiter & Schmidhuber, 1997) to predict the $\widehat{\mathbf{V}}^{unobs}$ on the basis of $\mathbf{V}^{obs}$,

$$\hat{v}_{i-1}^{unobs} = \Psi(\mathbf{V}^{obs}, \hat{v}_i^{unobs}; \psi), i = 1, 0, ..., -N + 2, \tag{4}$$

where $\hat{v}_i^{unobs}$ is the $i^{th}$ predicted unobserved latent feature representations of $\widehat{\mathbf{V}}^{unobs}$, and $\hat{v}_1^{unobs} = \mathbf{Mean}(\mathbf{V}^{obs})$. In order to reconstruct the latent feature representations, we use the smooth $L_1$ loss (Girshick, 2015) to optimize the $\mathcal{L}_{rec}$ as:

$$\mathcal{L}_{rec} = \sum_{i=-N+1}^{0} \delta(v_i^{unobs} - \hat{v}_i^{unobs}), \tag{5}$$

where $\delta$ is defined as:

$$\delta(v) = \begin{cases} 0.5v^2 & if\ ||v|| < 1 \\ ||v|| - 0.5 & otherwise, \end{cases} \tag{6}$$

where $||v||$ denotes the $l_1$ norm of $v$.

To further enhance the representation ability of the unobserved latent feature representations, we devise another self-supervised task. Specifically, we regard the feature pair $\{v_i^{unobs}, \hat{v}_i^{unobs}\}$ as the positive sample pair, $i = -N + 1, \cdots, 0$, and take $\{v_i^{unobs}, \hat{v}_j^{unobs}\}$ as the negative sample pair, $i \neq j$. After that, we present another self-supervised loss:

$$\mathcal{L}_{cts} = \sum_{i=-N+1}^{0} \sum_{j \neq i} \max(0, \delta(v_i^{unobs} - \hat{v}_i^{unobs}) - \delta(v_i^{unobs} - \hat{v}_j^{unobs}) + \Delta), \tag{7}$$

where $\Delta$ is a margin. It is worth noting that the first loss $\mathcal{L}_{rec}$ in (5) targets at reconstructing the latent feature representation $v_i$ as accurately as possible, while the second loss $\mathcal{L}_{cts}$ in (7) aims to minimize the discrepancy between the predicted unobserved feature representations and the corresponding ground-truth feature representations at each timestep, while it enlarges a margin $\Delta$ between the predicted unobserved and non-corresponding ground-truth feature representations. This can further assist in better reconstructing unobserved trajectories.

### 3.4 NOISE REDUNDANCY REDUCTION FORMER

Our Noise Redundancy Reduction Former (NRRFormer) module that is parameterized by $\Theta$ contains $L$ Noise Redundancy Reduction Blocks (NRRBlocks). Each NRRBlock attempts to filter out noise and redundancy in the predicted latent feature representations $\hat{\mathbf{V}}_l^{unobs}$, and integrate the resulting filtered feature representations and observed feature representations $\mathbf{V}^{obs}$ into a query embedding $\mathbf{Q}_{l+1}, l = 0, 1, \cdots, L-1$.

As shown in the Figure 3, The $l^{th}$ layer of NRRBlock takes as input the query embedding $\mathbf{Q}_l$ and the unobserved feature representations $\hat{\mathbf{V}}_l^{unobs}$ through a self-attention mechanism:

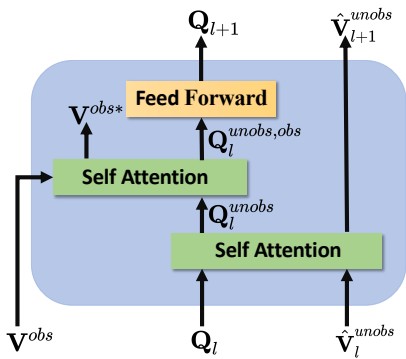

$$\mathbf{Q}_l^{unobs}, \hat{\mathbf{V}}_{l+1}^{unobs} = \mathbf{SelfAtt}(\mathbf{Q}_l || \hat{\mathbf{V}}_l^{unobs}; \theta_{l,1}), \quad (8)$$

where $||$ denotes the concatenation operation, the self-attention module is parameterized by $\theta_{l,1}$. $\mathbf{Q}_0$ is a random initialized tensor, $\hat{\mathbf{V}}_0^{unobs} = \hat{\mathbf{V}}^{unobs}$, and the $\mathbf{Q}_l^{unobs}$ represents the output query embedding. It is worth noting that the length of the query, denoted as $C$, is smaller than the length of $\hat{\mathbf{V}}_l^{unobs}$, denoted as $N$, so that information in $\hat{\mathbf{V}}_l^{unobs}$ is forced to condense and collate into the compact query embedding $\mathbf{Q}_l^{unobs}$, thereby filtering out redundancy and noise to extract the meaningful informa-

Figure 3: Structure of Noise Redundancy Reduction Block.

tion. After that, we utilize another self-attention module to integrate information of $\mathbf{V}^{obs}$ into the query embedding:

$$\mathbf{Q}_l^{unobs,obs}, \mathbf{V}^{obs*} = \mathbf{SelfAtt}(\mathbf{Q}_l^{unobs} || \mathbf{V}^{obs}; \theta_{l,2}), \quad (9)$$

where the self-attention module is parameterized by $\theta_{l,2}$, $\mathbf{Q}_l^{unobs,obs}$ represents the query embedding after integrating both the filtered unobserved trajectory features and the observed trajectory features. Through this self-attention operation, the information of $V^{obs}$ can be effectively distilled into Q, while enabling it to fuse with $V^{unobs}$, thereby facilitating the exchange of complementary information between them. Note that we assume the observed trajectory features $\mathbf{V}^{obs}$ do not contain noise or redundancy, because the features are obtained by encoding $\mathbf{X}^{obs}$. Therefore, the Equation (9) only integrates the information of $\mathbf{V}^{obs}$ into the query $\mathbf{Q}$ through self-attention, but not input the $\mathbf{V}^{obs*}$ into the next NRRBlock. At the end of the NRRBlock, we employ a feed forward layer to produce the query representation for the next layer,

$$\mathbf{Q}_{l+1} = \mathbf{FeedForward}(\mathbf{Q}_l^{unobs,obs}; \theta_{l,3}), \quad (10)$$

where the feed forward layer is parameterized by $\theta_{l,3}$. We utilize $L$ NRRBlocks to denoise and reduce redundancy in the unobserved trajectory features while effectively fusing the observed trajectory features. Finally we utilize $\mathbf{Q}_L$ for future trajectory prediction:

$$\{\widehat{\mathbf{X}}^k\}_{k \in [0, K-1]} = \Omega(\mathbf{Q}^L; \omega), \quad (11)$$

where $\Omega$ represents the decoder module parameterized by $\omega$. The decoder module can be the same structure as in previous trajectory prediction models (Zhou et al., 2022; Liang et al., 2020), enabling our method to be generalizable.

### 3.5 OPTIMIZATION AND INFERENCE

We adopt the commonly used winner-takes-all strategy (Zhao et al., 2021) on the obtained $K$ multi-modal trajectories $\{\widehat{\mathbf{X}}^k\}_{k \in [0, K-1]}$, which regresses the trajectory closest to the ground truth, denoted as $\mathcal{L}_{reg}$. In order to help the downstream planner make better decisions, a classification loss $\mathcal{L}_{cls}$ is also adopted to score each trajectory. Here, we adopt the same $\mathcal{L}_{reg}$ and $\mathcal{L}_{cls}$ as those in the corresponding backbones (see Appendix A.5 for details of $\mathcal{L}_{reg}$ and $\mathcal{L}_{cls}$). Finally, the total loss function can be expressed as:

$$\mathcal{L} = \mathcal{L}_{reg} + \mathcal{L}_{cls} + \alpha \mathcal{L}_{rec} + \beta \mathcal{L}_{cts}, \quad (12)$$

where $\alpha$ and $\beta$ are three trade-off hyper-parameters. We provide the pseudo-code of our training procedure in Appendix A.1.

For inference, when only 2 observed trajectory points of a target vehicle are collected, we first extract the latent feature representations based on the backbone $\Phi$, and then apply our backward forecasting mechanism to predict the latent feature representations of previous $N$ unobserved locations of the target agent by the networks $\Psi$. After that, the NRRFormer $\Theta = \{\theta_{l,1}, \theta_{l,2}, \theta_{l,3}\}_{l=1}^{L}$ filters out the noise and redundancy in the unobserved latent feature representations and integrates the filtered features and observed latent feature representations into query embedding. Finally, the query embedding are fed into the decoder network $\Omega$ for instantaneous trajectory prediction.

## 4 EXPERIMENTS

### 4.1 DATASETS

We evaluate our method for the instantaneous trajectory prediction tasks on two widely used benchmark datasets, Argoverse (Chang et al., 2019) and NuScene (Caesar et al., 2020).

**Argoverse Datasets:** This dataset contains a total of 324,557 scenes, which are split into 205,492 training scenes, 39,472 validation scenes, and 78,143 testing scenes. The observation duration for both the training and validation sets is 5 seconds with a sampling frequency of 10Hz. In contrast to previous approaches taking the first 2 seconds (i.e., 20 locations) as the observed trajectory and the last 3 seconds as the future ground-truth trajectory, we only utilize 2 observed locations, and predict the future trajectory of the last 3 seconds in our experiments.

**NuScene Datasets** The dataset consists of 32,186 training, 8,560 validation, and 9,041 test samples. Each sample is a sequence of x-y coordinates with a duration of 8 seconds and a sample frequency of 2Hz. Previous approaches usually take the first 2 seconds (i.e., 5 locations) as the observed trajectory and the last 6 seconds as the future ground-truth trajectory. However, we leverage only 2 observed locations to predict the future trajectory of the last 6 seconds in the experiments.

### 4.2 IMPLEMENTATION DETAILS

We perform the experiments using two different backbone models, HiVT (Zhou et al., 2022) and LaneGCN (Liang et al., 2020). Specifically, we utilize the temporal encoder in HiVT and the ActorNet in LaneGCN to extract the latent feature representations, respectively. We set the feature dimensions $d$ to 64 and 128 when using HiVT and LaneGCN as the backbone, respectively. The hidden size of the LSTM for predicting unobserved latent feature representations is set to $d$. The NRRFormer consists of three NRRBlocks. In our experiments, the predicted unobserved length $N$ is set to 10 for the Argoverse dataset and 4 for the nuScenes dataset, and correspondingly, we set the query embedding length to $C = 4$ for the Argoverse dataset and $C = 2$ for the nuScenes dataset. In addition, we set the trade-off hyper-parameters $\alpha$ and $\beta$ to 0.1 and 0.1.

### 4.3 BASELINES AND EVALUATION METRICS

We first compare with two most related works: MOE (Sun et al., 2022a) and Distill (Monti et al., 2022). Since we use HiVT (Zhou et al., 2022) and LaneGCN (Liang et al., 2020) as our backbone, respectively, we also compare our method with them. When using HiVT as the backbone, we denote our method as ITPNet+HiVT. When using LaneGCN as the backbone, we denote our method as ITPNet+LaneGCN. To evaluate these methods, we employ three popular evaluation metrics (Zhao et al., 2021; Gu et al., 2021; Wang et al., 2022b), minADE@$K$, minFDE@$K$, and minMR@$K$, where $K$ represents the number of the generated trajectories. we set $K$ to 1 and 6 in our experiments.

### 4.4 RESULT AND ANALYSIS

**Performance on Instantaneous Trajectory Prediction**: To demonstrate the effectiveness of our method for instantaneous trajectory prediction, we compare our method with the state-of-the-art baselines. The results are listed in Table 1. Based on Table 1, ITPNet+LaneGCN and ITPNet+HiVT significantly outperforms LaneGCN and HiVT, respectively. This illustrates current state-of-the-art

Table 1: minADE@$K$, minFDE@$K$, and MR@$K$ of different methods on Argoverse and nuScenes, respectively.

| Dataset | Methods | K=1 | | | K=6 | | |
|---|---|---|---|---|---|---|---|
| | | minADE | minFDE | minMR | minADE | minFDE | minMR |
| Argoverse | LaneGCN (Liang et al., 2020) | 4.204 | 8.647 | 0.861 | 1.126 | 1.821 | 0.278 |
| | HiVT (Zhou et al., 2022) | 4.158 | 8.368 | 0.846 | 1.085 | 1.712 | 0.249 |
| | MOE (Sun et al., 2022a) | 3.312 | 6.840 | 0.794 | 0.939 | 1.413 | 0.177 |
| | Distill (Monti et al., 2022) | 3.251 | 6.638 | 0.771 | 0.968 | 1.502 | 0.185 |
| | ITPNet+LaneGCN | 2.922 | 5.627 | 0.765 | 0.894 | 1.425 | 0.173 |
| | ITPNet+HiVT | **2.631** | **5.703** | **0.757** | **0.819** | **1.218** | **0.141** |
| nuScenes | LaneGCN (Liang et al., 2020) | 6.125 | 14.300 | 0.935 | 1.878 | 3.497 | 0.630 |
| | HiVT (Zhou et al., 2022) | 6.564 | 13.745 | 0.914 | 1.772 | 2.836 | 0.505 |
| | MOE (Sun et al., 2022a) | 5.705 | 12.619 | 0.913 | 1.712 | 2.813 | 0.494 |
| | Distill (Monti et al., 2022) | 5.950 | 12.606 | 0.911 | 1.759 | 2.861 | 0.483 |
| | ITPNet+LaneGCN | 5.739 | 13.555 | 0.919 | 1.679 | 3.146 | 0.580 |
| | ITPNet+HiVT | **5.514** | **12.584** | **0.909** | **1.503** | **2.628** | **0.483** |

Table 2: Ablation study of our method for $\mathcal{L}_{rank}$, $\mathcal{L}_{rec}$ and $\mathcal{L}_{cts}$ on the Argoverse dataset.

| $\mathcal{L}_{rec}$ | $\mathcal{L}_{ctx}$ | NRRFormer | K=1 | | | K=6 | | |
|---|---|---|---|---|---|---|---|---|
| | | | minADE | minFDE | minMR | minADE | minFDE | minMR |
| | | | 4.158 | 8.368 | 0.846 | 1.085 | 1.712 | 0.249 |
| ✓ | | | 2.646 | 5.790 | 0.763 | 0.841 | 1.285 | 0.154 |
| ✓ | ✓ | | **2.615** | 5.733 | 0.761 | 0.832 | 1.262 | 0.149 |
| ✓ | ✓ | ✓ | 2.631 | **5.703** | **0.757** | **0.819** | **1.218** | **0.141** |

trajectory prediction approaches cannot well handle the case of instantaneous observed trajectories. However, when plugging our framework into these two models, respectively, the performance is significantly improved. This shows our method is effective for instantaneous trajectory prediction, and is compatible with different trajectory prediction models. Moreover, our methods achieve better performance than MOE and Distill, which indicates the effectiveness of our methods once more.

**Ablation Study**: We conduct ablation studies on the Argoverse dataset, and we employ HiVT (Zhou et al., 2022) as the backbone. Table 2 shows the results. When the $\mathcal{L}_{rec}$ is applied to the loss function, our method significantly improves the performance. This indicates the effectiveness of our proposed backward forecast mechanism for predicting the latent feature representations of previous unobserved trajectories. The loss $\mathcal{L}_{cts}$ further boosts the performance of the model, demonstrating the self-supervised task is meaningful. Moreover, our method can further improve the performance when integrating our NRRFormer, underscoring the effectiveness of our NRRFormer in filtering out noise and redundancy from the predicted unobserved latent features.

**Analysis of Different Lengths $N$**: We investigate the influence of different lengths $N$ of unobserved trajectories on instantaneous trajectory prediction. We use HiVT as the backbone on the Argoverse dataset. The results are listed in Table 3. Note that when NRRFormer is not used, we directly concatenate the predicted unobserved features with observed features for future trajectory prediction. As $N$ increases, the performance of the model is gradually improved. This reveals that predicting more latent feature representations can introduce more useful information, and thus be beneficial to trajectory prediction. However, when $N$ exceeds a certain value ($N > 3$), the performance deteriorates. This is attributed to the introduction of noise and redundancy when predicting a longer feature sequence. When applying our NRRFormer, we perform our method directly using a bigger value of $N$ (e.g., $N = 10$), without tuning it carefully. Our method with NRRFormer achieves the best results in Table 3. We believe if we tune $N$ carefully, the performance of our method can be further improved. This further indicates the superiority of our NRRFormer.

**Qualitative Results**: We perform a visualization of the predicted multi-modal trajectories generated by MOE, Distill, HiVT, and our proposed method ITPNet+HiVT respectively on Argoverse dataset with only 2 observed locations. The results are shown in Figure 4. We observe that our method exhibits diversity and more accurate trajectory prediction than other baselines in the scenario of

Table 3: Analysis of backward forecasting with different $N$ and effectiveness of NRRFormer on Argoverse.

| NRRFormer | N | K=1 | | | K=6 | | |
|---|---|---|---|---|---|---|---|
| | | minADE | minFDE | minMR | minADE | minFDE | minMR |
| ✗ | | 4.019 | 8.108 | 0.829 | 1.068 | 1.678 | 0.241 |
| ✗ | 1 | 3.458 | 7.112 | 0.801 | 0.969 | 1.494 | 0.193 |
| ✗ | 2 | 2.898 | 6.024 | 0.777 | 0.872 | 1.329 | 0.160 |
| ✗ | 3 | **2.615** | 5.733 | 0.761 | 0.832 | 1.262 | 0.149 |
| ✗ | 4 | 2.632 | 5.803 | 0.762 | 0.845 | 1.291 | 0.154 |
| ✗ | 6 | 2.709 | 5.895 | 0.769 | 0.867 | 1.302 | 0.161 |
| ✗ | 8 | 3.045 | 6.231 | 0.789 | 0.903 | 1.410 | 0.181 |
| ✗ | 10 | 3.394 | 6.956 | 0.801 | 0.967 | 1.522 | 0.196 |
| ✓ | 10 | 2.631 | **5.703** | **0.757** | **0.819** | **1.218** | **0.141** |

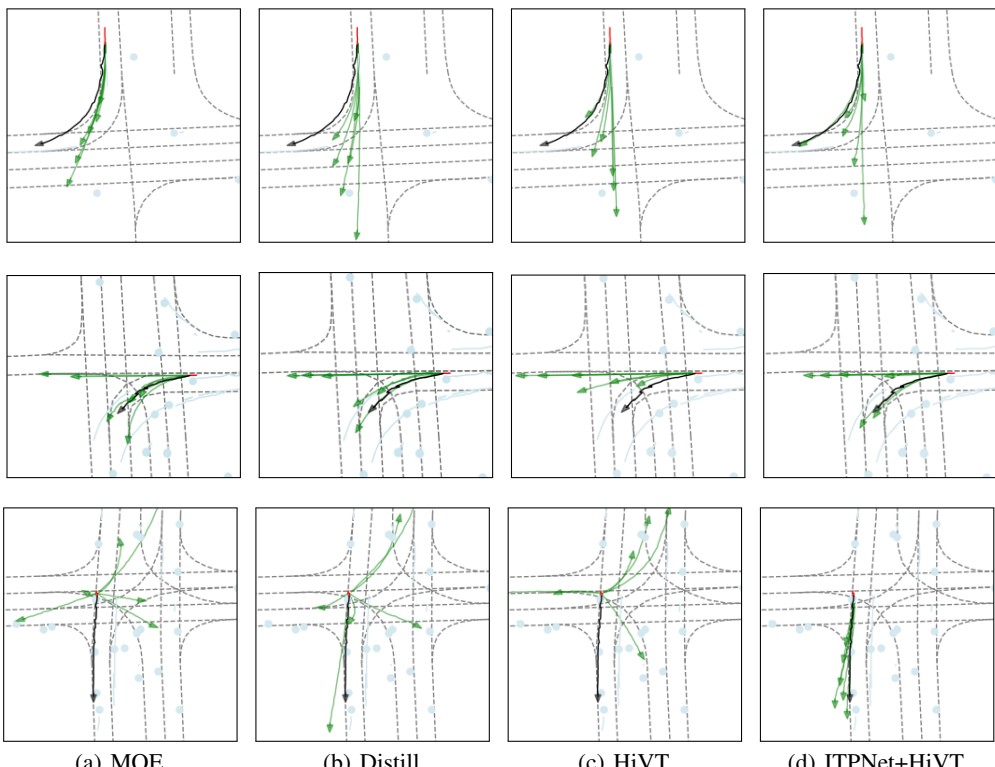

|  (a) MOE  |  (b) Distill  |  (c) HiVT  |  (d) ITPNet+HiVT  |

Figure 4: Qualitative results of a) MOE, b) Distill, c) HiVT, d) ITPNet+HiVT on Argoverse. The tracked trajectories are shown in red, the ground-truth trajectories are shown in black, and the predicted multi-modal trajectories are shown in green.

turning and going straight. This suggests that our method can handle different driving scenarios and can achieve improved predictions with only 2 tracked locations.

## 5 CONCLUSION

In this paper, we investigated a challenging problem of instantaneous trajectory prediction with very few tracked locations. We proposed a plug-and-play approach that backwardly predicted the latent feature representations of unobserved locations, to mitigate the issue of the lack of information. Considering the noise and redundancy in unobserved feature representations, we designed the NR-RFormer to remove them and integrate the resulting filtered features and observed trajectory features into a compact query embedding for future trajectory prediction. Extensive experimental results demonstrated that the proposed method can be effective for instantaneous trajectory prediction, and can be compatible with different trajectory prediction models.

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

# A  APPENDIX

## A.1  TRAINING ALGORITHM OF ITPNET

We present the pesudo code of training ITPNet in Algorithm 1

---

**Algorithm 1:** Training Procedure of ITPNet

---

**Input:** input trajectory $\mathbf{X} = \{\mathbf{X}^{obs}, \mathbf{X}^{unobs}\}$, ground-truth trajectory $\mathbf{X}^{gt}$, query embedding $\mathbf{Q}$, layers $L$ of NRRFormer, trade-off hyper-parameters: $\alpha$, and $\beta$.
**Output:** Network parameters: $\phi$, $\psi$, $\{\theta_{l,1}, \theta_{l,2}, \theta_{l,3}\}_{l=1}^{L}$, and $\omega$.
**Initialize:** Randomly initialize $\phi$, $\psi$, $\{\theta_{l,1}, \theta_{l,2}, \theta_{l,3}\}_{l=1}^{L}$, $\omega$, and $\mathbf{Q}$.
**while** *not converges* **do**

> Compute latent feature representations $\mathbf{V}^{obs} = \Phi(\mathbf{X}^{obs}; \phi)$ and $\mathbf{V}^{unobs} = \Phi(\mathbf{X}^{unobs}; \phi)$;
> Backward forecast $\widehat{\mathbf{V}}^{unobs}$ by $\widehat{\mathbf{V}}^{unobs} = \Psi(\mathbf{V}^{obs}; \psi)$;
> Compute $\mathcal{L}_{rec}, \mathcal{L}_{cts}$ by Eq. (5) and (7), respectively;
> Employ NRRFormer to filter out redundancy and noise in predicted unobserved latent feature representations and integrate the resulting filtered feature representations and observed feature representations into $Q$, by
> $\hat{\mathbf{V}}_0^{unobs} = \hat{\mathbf{V}}^{unobs}$;
> **for** $l = 0...L-1$ **do**
>
> > $\mathbf{Q}_l^{unobs}, \hat{\mathbf{V}}_{l+1}^{unobs} = \mathbf{SelfAtt}(\mathbf{Q}_l || \hat{\mathbf{V}}_l^{unobs}; \theta_{l,1})$ ;
> > $\mathbf{Q}_l^{unobs,obs}, \mathbf{V}^{obs*} = \mathbf{SelfAtt}(\mathbf{Q}_l^{unobs} || \mathbf{V}^{obs}; \theta_{l,2})$;
> > $\mathbf{Q}_{l+1} = \mathbf{FeedForward}(\mathbf{Q}_l^{unobs,obs}; \theta_{l,3})$;
>
> **end**
> Predict trajectory $\{\widehat{\mathbf{X}}^k\}_{k \in [0,K-1]} = \Omega(\mathbf{Q}_L; \omega)$;
> Compute $\mathcal{L}_{reg}, \mathcal{L}_{cls}$ through $\{\widehat{\mathbf{X}}^k\}_{k \in [0,K-1]}$;
> Calculate the total loss $\mathcal{L}$ by $\mathcal{L} = \mathcal{L}_{reg} + \mathcal{L}_{cls} + \alpha \mathcal{L}_{rec} + \beta \mathcal{L}_{cts}$;
> Update model parameters $\phi$, $\psi$, $\{\theta_{l,1}, \theta_{l,2}, \theta_{l,3}\}_{l=0}^{L-1}$, $\omega$ and query embedding $\mathbf{Q}$ by minimizing $\mathcal{L}$.

**end**

---

## A.2  RESULTS WITH DIFFERENT LENGTHS OF OBSERVATIONS

To further demonstrate the effectiveness of our method, we perform HiVT and LaneGCN with different lengths of observed locations $T$. Table 4 reports the results. One interesting point is that our ITPNet+LaneGCN with $T = 2$ achieves comparable performance to LaneGCN with $T = 5$ when $K = 1$ on the nuScenes dataset. This means that our method can averagely save 1.5 seconds for trajectory prediction, compared to LaneGCN. If a car has a driving speed of 70 kilometers per hour on an urban road, our method can save around 30 meters to observe the agent for trajectory prediction, compared to LaneGCN.

## A.3  CONVERGENCE ANALYSIS

We study the convergence of our method on Argoverse and nuScenes. The curves of the total loss of our method are shown in Figure 5. we can see the loss decreases as the training steps, and it finally levels off.

## A.4  FAILURE CASES OF ITPNET

We provide failure cases of ITPNet+HiVT on Argoverse dataset, as shown in Figure 6. The model fails (1) when the future intention of the agents suddenly changes (a, d); (2) the future behavior is complex and hard to perceive from observed trajectories, such as overtaking; (3) the agent does not follow the traffic rules, such as turning left from the lane for right turns (c).

Table 4: minADE@$K$, minFDE@$K$, and MR@$K$ of methods with different observed locations ($T$) on Argoverse and nuScenes, respectively.

| Dataset | Method | T | K=1 minADE | minFDE | minMR | K=6 minADE | minFDE | minMR |
|---|---|---|---|---|---|---|---|---|
| Argoverse | HiVT (Zhou et al., 2022) | 3 | 2.958 | 6.601 | 0.816 | 0.930 | 1.502 | 0.190 |
| | | 4 | 2.777 | 5.895 | 0.766 | 0.852 | 1.287 | 0.152 |
| | | 5 | 2.510 | 5.523 | 0.747 | 0.809 | 1.203 | 0.137 |
| | | 20 | 2.032 | 4.579 | 0.691 | 0.698 | 1.053 | 0.107 |
| | ITPNet+HiVT | 2 | 2.631 | 5.703 | 0.757 | 0.819 | 1.218 | 0.141 |
| | LaneGCN (Liang et al., 2020) | 3 | 3.512 | 7.607 | 0.837 | 1.007 | 1.642 | 0.234 |
| | | 4 | 3.093 | 6.805 | 0.817 | 0.941 | 1.520 | 0.202 |
| | | 5 | 2.817 | 6.401 | 0.804 | 0.878 | 1.417 | 0.171 |
| | | 20 | 2.248 | 5.209 | 0.746 | 0.788 | 1.191 | 0.129 |
| | ITPNet+LaneGCN | 2 | 2.922 | 5.627 | 0.765 | 0.894 | 1.425 | 0.173 |
| nuScenes | HiVT (Zhou et al., 2022) | 2 | 6.564 | 13.745 | 0.914 | 1.772 | 2.836 | 0.505 |
| | | 3 | 5.182 | 11.887 | 0.908 | 1.455 | 2.564 | 0.445 |
| | | 4 | 5.159 | 11.836 | 0.903 | 1.442 | 2.567 | 0.442 |
| | | 5 | 5.002 | 11.520 | 0.899 | 1.431 | 2.559 | 0.419 |
| | ITPNet+HiVT | 2 | 5.514 | 12.584 | 0.909 | 1.503 | 2.628 | 0.483 |
| | LaneGCN (Liang et al., 2020) | 2 | 6.125 | 14.300 | 0.935 | 1.878 | 3.497 | 0.630 |
| | | 3 | 5.853 | 13.845 | 0.941 | 1.697 | 3.160 | 0.592 |
| | | 4 | 5.708 | 13.492 | 0.933 | 1.653 | 3.060 | 0.569 |
| | | 5 | 5.663 | 13.427 | 0.934 | 1.647 | 3.052 | 0.573 |
| | ITPNet+LaneGCN | 2 | 5.739 | 13.555 | 0.919 | 1.679 | 3.146 | 0.580 |

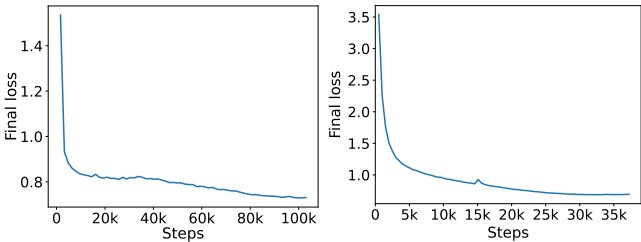

Figure 5: Convergence analysis of our method. Left for Argoverse and right for nuScenes.

## A.5 DETAILS ABOUT $\mathcal{L}_{reg}$ AND $\mathcal{L}_{cls}$ OF BACKBONES

**HiVT** parameterizes the distribution of future trajectories as a mixture model where each mixture component is a Laplace distribution. The regression loss $\mathcal{L}_{reg}$ is defined as:

$$\mathcal{L}_{reg} = \sum_{i=3}^{M+2} \log \frac{1}{2b} \exp(-\frac{|\hat{x}_i - \mu_i|}{b}), \tag{13}$$

where $b$ is a learnable scale parameter of Laplace distribution, $\hat{x}^i$ is the predicted future trajectory closest to the ground-truth future trajectory and $\mu_i$ is the ground-truth future trajectory. The $\mathcal{L}_{cls}$ is defined as cross-entropy loss to optimize the mixing coefficients,

$$\mathcal{L}_{cls} = -\sum_{k=1}^{K} \pi^k \log \hat{\pi}^k, \tag{14}$$

where $\pi^k$ and $\hat{\pi}^k$ are the probability of the $k^{th}$ trajectory to be selected, and $\pi^k = 1$ if and only if $\hat{\mathbf{X}}^k$ is the predicted future trajectory closest to the ground-truth future trajectory.

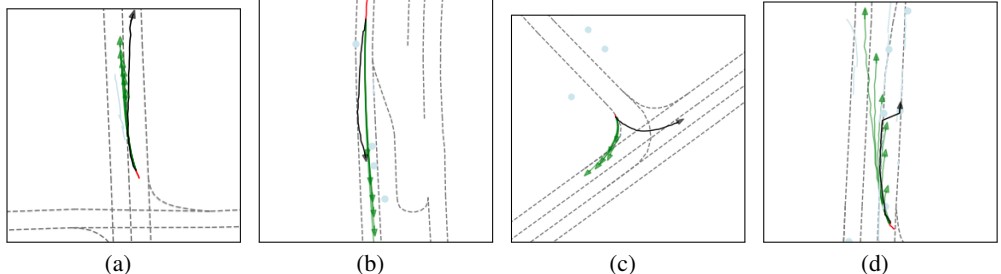

| (a) | (b) | (c) | (d) |

Figure 6: Failure case of ITPNet+HiVT on Argoverse. The observed trajectories are shown in red, the ground-truth trajectories are shown in black, and the predicted multi-modal trajectories are shown in green.

**LaneGCN** employ smooth $L_1$ loss as $\mathcal{L}_{reg}$, which is defined as,

$$\mathcal{L}_{reg} = \sum_{i=3}^{M+2} \delta(\hat{x}_i - x_i), \tag{15}$$

where the definition of $\delta$ is same as Equation (6). The LaneGCN employs max-margin loss as $\mathcal{L}_{cls}$, which is defined as,

$$\mathcal{L}_{cls} = \frac{1}{K-1} \sum_{k \neq k'} \max(0, \pi^k + \epsilon - \pi^{k'}), \tag{16}$$

where the $k^{th}$ predicted future trajectory is the closest one to the ground-truth future trajectory. This max-margin loss pushes the closest one away from others at least by a margin $\epsilon$.

### A.6 IMPLMENTATIONS OF BASELINES

**MOE.** To have a fair comparison, we extend the HiVT backbones used in this paper to MOE. We use the A-A Interaction and A-L Interaction in HiVT to as the In-patch Aggregation in MOE and replace the Global interaction in HiVT to replace the Cross-patch Aggregation in MOE. In addition, followed by MOE, we employ the soft-pretraining with masked trajectory complement and Context restoration tasks.

**Distill.** We also utilize HiVT as the backbone of Distill for fair comparison. In addition, followed by Distill we distill knowledge from the output of the encoder (output of Global Interaction in HiVT) and decoder (output of last hidden layer).

