# OpenReview forum: "ITPNet: Towards Instantaneous Trajectory Prediction for Autonomous Driving"
_ICLR.cc/2024/Conference — Submitted to ICLR 2024_

### Official Review · Reviewer_ptje · 2023-10-15

**Soundness:** 3 good
**Presentation:** 2 fair
**Contribution:** 3 good
**Rating:** 8
**Confidence:** 5

**Summary:**

The authors identified an important limitation of the traditional trajectory prediction approaches that they require 2 seconds of observations to make accurate predictions. They took the HiVT model as an example, and its prediction performance drops significantly when only two observations are available. However, for an autonomous driving vehicle to safely operate, a trajectory prediction model needs to be able to make accurate predictions for an agent before it has been observed for 2 seconds.

To tackle this limitation, the authors proposed an instantaneous trajectory prediction approach, called ITPNet. ITPNet is able to make predictions with only two observations.

The key idea of ITPNet is to use a backward forecasting module to reconstruct the unobserved latent feature representations of the agent using the two observed ones.

The authors also proposed a Noise Redundancy Reduction Former (NRRFormer) module to filter the reconstructed unobserved features.

ITPNet is a generic plug-and-play approach that can be used in combination with any trajectory prediction backbones. In this paper, the authors applied ITPNet on HiVT and LaneGCN backbones. They evaluated the resulting ITPNet+HiVT and ITPNet+LaneGCN models on the Argoverse and nuScenes datasets. The evaluation results show that, when using two observations, ITPNet significantly improves the prediction performance over the HiVT and LaneGCN baselines.

The authors also performed ablation studies to evaluate the contributions from the reconstruction loss and NRRFormer.

**Strengths:**

* I like the motivation of this paper. It attempts to tackle an important limitation of the traditional trajectory prediction approaches.

* ITPNet is a generic plug-and-play approach that can be used in combination with any trajectory prediction backbones. The authors applied applied ITPNet on two popular open-sourced backbones, HiVT and LaneGCN. This makes it a lot easier for other people to adopt this work.

* The result shows ITPNet significantly improves the prediction performance over the HiVT and LaneGCN baselines when using two observations.

* From author's response during the rebuttal, I now understand that the model is able to make predictions using all the available observed history, which makes it a practical solution for a real-world autonomous driving system.

**Weaknesses:**

* From the method and evaluation sections of the paper, it's not very clear whether this method is able to make predictions using all the available observed history. In the method section, it will be useful to clarify this and explain how this method is able to do so. In the evaluation section, it will be useful to make a curve plot to compare the prediction performances when different lengths of observed history are available. To match a real-world deployed prediction system, you should only have one ITPNet+HiVT model and do predictions with different lengths of available history. It will also be useful to make a curve for the HiVT baseline model as well.

* The ablation study result was incomplete in the original submission, but I am good with the additional results provided in the rebuttal.

**Questions:**

* Is ITPNet able to able to adaptively adjust the length of history used for different agents?

* Will N=3 with NRRFormer enabled yield better result?

---

> ### Author Response · Authors · 2023-11-19
> **Response to Reviewer ptje [1/2]**
>
> > Q1:  From the evaluation, it looks like ITPNet is not able to adaptively adjust the length of history used for different agents. For example, when an agent has 20 observations available, ITPNet still only makes predictions with two observations, and the prediction accuracy is lower than the HiVT baseline that uses 20 observations. In practice, people would like to use as many available observations as possible to make the most accurate predictions for an agent.
>
> R1: Thanks for raising this concern which helps us to clarify the problem setting. We guess this concern is caused by the misunderstanding on the length of the history trajectories. We would like to clarify that ITPNet is actually able to adaptively adjust the length of history used for different agents. This adaptability is achieved through the utilization of both LSTM in the backward forecasting module and the self-attention mechanism in the NRRFormer module. These components enables our method to handle the input historical trajectories with varying lengths.
>
> Besides, please kindly note that our problem setting is the same as that of MOE [1], the first work in instantaneous trajectory prediction. Thus, for a fair comparison, we also focus on the most extreme scenario, where only 2 frames of locations can be observed.
>
> [1] Human Trajectory Prediction with Momentary Observation. CVPR'22
>
> >Q2: The ablation result suggests that this is still an unfinished work. According to Table 3, without NRRFormer, ITPNet has better performance using N=3 than using N=10. However, when NRRFormer is enabled, the authors just picked N=10 without any tuning, and the paper says they don't know how the performance of N=3 will be when NRRFormer is enabled.
>
> R2: Thanks for your advice. Due to the deadline, we directly set $N=10$ as the length of predicted unobserved locations without tuning. Following your suggestion, we conducted experiments with more $N$. The experimental results are listed in the table below. We will add them in the final version.
>
> |  |           | | K=6| | | |K=6 | |
> |- |-          |-|-|-|-|-|-|-|
> | N| NRRFormer | minADE |minFDE | minMR| NRRFormer| minADE| minFDE| minMR|
> |0 |    ×      |1.068|1.678|0.241|-|-|-|-|
> |1 |    ×      |0.969|1.494|0.193|√|0.964|1.498|0.194|
> |2 |    ×      |0.872|1.329|0.160|√|0.868|1.323|0.158|
> |3 |    ×      |0.832|1.262|0.149|√|0.828|1.254|0.147|
> |4 |    ×      |0.845|1.291|0.154|√|0.824|1.240|0.146|
> |5 |    ×      |0.859|1.312|0.156|√|0.822|1.232|0.145|
> |6 |    ×      |0.867|1.302|0.161|√|0.823|1.231|0.145|
> |7 |    ×      |0.881|1.375|0.173|√|0.820|1.222|0.143|
> |8 |    ×      |0.903|1.410|0.181|√|0.821|1.222|0.142|
> |9 |    ×      |0.933|1.453|0.187|√|0.819|1.220|0.142|
> |10|    ×      |0.967|1.522|0.196|√|0.819|1.218|0.141|
>
>
> Based on the above table, we have some interesting findings:
> When NRRFormer is not enabled, the prediction error initially decreases and then increases with the increase of $N$. This is attributed to the introduction of noise and redundancy when predicting a longer feature sequence. When NRRFormer is enabled, the performance of our method is consistently improved as $N$ increases. This illustrates our NRRFormer model indeed can filter out redundant and noisy information, demonstrating its effectiveness.

---

> > ### Comment · Reviewer_ptje · 2023-11-20
> > **Thank you for clarifying the dynamic history length and ablation study**
> >
> > Thank you for clarifying the dynamic history length and ablation study. I have updated my score.
> >
> > Please address this comment in the final version:
> > > From the method and evaluation sections of the paper, it's not very clear whether this method is able to make predictions using all the available observed history. In the method section, it will be useful to clarify this and explain how this method is able to do so. In the evaluation section, it will be useful to make a curve plot to compare the prediction performances when different lengths of observed history are available. To match a real-world deployed prediction system, you should only have one ITPNet+HiVT model and do predictions with different lengths of available history. It will also be useful to make a curve for the HiVT baseline model as well.

---

> > > ### Author Response · Authors · 2023-11-20
> > > **Re: Thank you for clarifying the dynamic history length and ablation study**
> > >
> > > Dear Reviewer ptje，
> > >
> > > Thank you very much! We will incorporate your suggestion to enhance the final version of our work.
> > >
> > > Best Regards, Authors

---

> ### Author Response · Authors · 2023-11-19
> **Response to Reviewer ptje [2/2]**
>
> > Q3: From Table 2, NRRFormer barely provides any performance boost.
>
> R3: Thanks for your comments.  We would like to kindly remind you that the NRRFormer module is effective for the instantaneous trajectory prediction task by the following three aspects:
>
> First, based on the table listed in the response to Q2,  we find when NRRFormer is disabled, the prediction error initially decreases and then increases with the increase of $N$. This phenomenon can be attributed to the introduction of noise and redundancy when predicting a longer feature sequence. Nevertheless, with the activation of NRRFormer, our method exhibits consistent improvement in performance as $N$ increases. For example,  NRRFormer with $N=10$ significantly enhances performance compared to the method without NRRFormer ($N=10$). Therefore, this illustrates that our NRRFormer model can indeed filter out redundant and noisy information, showcasing its effectiveness.
>
> Second, please kindly note that if we do not predict unobserved history trajectories based on the observed trajectories, our method will become the backbone. Thus, the performance of the backbone with $T+N$ observed points should serve as the upper bound for our method that involves using $T$ observed points and predicting $N$ unobserved historical points. Let's take $T=2$ and $N=3$ as an example. As listed in Table 4 in the Appendix, when using $T+N=5$ observed points, the scores minADE(K=6), minFDE(K=6) and minMR(K=6) of HiVT are 0.809, 1.203 and 0.137 on the Argoverse dataset, respectively.  We loosely consider these scores to be an upper bound for our method with $T=2$ and $N=3$.
> When NRRFormer is not enabled, the scores minADE(K=6), minFDE(K=6), and minMR(K=6) of our method with $T=2$ and $N=3$ are 0.832, 1.262 and 0.149 respectively, as listed in Table 2. The improvement margins for reaching the upper bound are $0.832-0.809=0.023$ for minADE(K=6), $1.262-1.203=0.059$ for minFDE(K=6), $0.149-0.137=0.012$ for minMR(K=6), respectively.
> When NRRFormer is enabled, the minADE(K=6), minFDE(K=6) and minMR(K=6) values for our method with $T=2$ and $N=3$ are 0.828, 1.254 and 0.147, respectively. The performance gains within the improvement margins are $\frac{(0.832-0.828)}{0.023}=17.4\\%$ for minADE(K=6), $\frac{(1.262-1.254)}{0.059}=13.6\\%$ for minFDE(K=6) and $\frac{(0.149-0.147)}{0.012}=16.7\\%$ for minMR(K=6), respectively.
>
> Third, in order to verify that the improvement of NRRFormer is statistically significant, we further run the experiments 4 times and perform the t-test. The experimental results are listed in the table below. Almost all p-values are less than the predetermined significance level of $\alpha=0.05$. This suggests that the performance is significantly improved, i.e., all the components in our method are effective for the instantaneous trajectory prediction task.
>
> |           |           |          |      |       | K=1  |       |     |       |
> |-          |-          |-         |-     |-      |-     |-      |-    | -     |
> | $L_{rec}$ | $L_{ctx}$ | NRRFormer|minADE|p-value|minFDE|p-value|minMR|p-value|
> |          |           |          |   $4.146±0.080$   |   -    |   $8.347±0.074$   |   -    |  $0.844±0.011$   |   -    |
> | √          |       |  |    $2.667±0.056$       |     $8.9\times10^{-6}$     |   $5.801±0.054$   |   $9.3\times10^{-7}$    |  $0.764±0.002$   |   $2.1\times10^{-4}$    |
> | √         | √         |          |  $2.615±0.013$    |    $0.049$   |   $5.737±0.016$   |    $0.030$   |  $0.761±0.001$   |   $0.029$    |
> | √         | √         |     √    |   $2.577±0.041$   |   $0.065$   |   $5.646±0.051$   |   $0.014$    |  $0.753±0.003$   |   $0.007$    |
>
>
>
> |           |           |          |      |       | K=6  |       |     |       |
> |-          |-          |-         |-     |-      |-     |-      |-    | -     |
> | $L_{rec}$ | $L_{ctx}$ | NRRFormer|minADE|p-value|minFDE|p-value|minMR|p-value|
> |          |           |          |   $1.089±0.003$   |   -    |   $1.719±0.006$   |   -    |  $0.248±0.002$   |   -    |
> | √          |       |  |    $0.843±0.003$       |     $2.4\times10^{-7}$     |   $1.293±0.007$   |   $5.0\times10^{-9}$    |  $0.155±0.002$   |   $1.9\times10^{-6}$    |
> | √         | √         |          |  $0.832±0.002$    |    $0.005$   |   $1.263±0.003$   |    $0.002$   |  $0.149±0.001$   |   $8.1\times10^{-4}$    |
> | √         | √         |     √    |   $0.822±0.002$   |   $9.4\times10^{-4}$   |   $1.222±0.003$   |   $2.0\times10^{-4}$    |  $0.143±0.001$   |   $3.9\times10^{-4}$    |

---

> > ### Comment · Reviewer_1F4C · 2023-11-20
> >
> > Thank you for replying to some of the reviewer questions.
> >
> > Can you clarify if these repeated trials are over different random seeds?

---

> > > ### Author Response · Authors · 2023-11-20
> > > **Trials with different random seeds**
> > >
> > > Dear Reviewer 1F4C,
> > >
> > > Thanks for your comment. Yes, we performed the repeated trials with different random seeds. We are making every effort to promptly address all reviewer questions and submit the responses as soon as possible. We eagerly anticipate engaging in discussions with the reviewers. We are confident that your valuable comments will significantly contribute to enhancing the quality of our work.
> > >
> > > Best Regards, Authors

---

### Official Review · Reviewer_LWjU · 2023-10-28

**Soundness:** 4 excellent
**Presentation:** 4 excellent
**Contribution:** 3 good
**Rating:** 8
**Confidence:** 3

**Summary:**

The paper proposes a plug-and-play approach for instantaneous trajectory prediction when there are only two observations.
The proposed ITPNet considers the lack of information as the reason for poor prediction when there are few observations, and uses backwardly prediction to predict unobserved representation as complementary information.
The authors discovered that as this additional information increases, the amount of information increases, but the quality deteriorates.
Therefore, they proposed an NRRFormer that can filter this.
The proposed method significantly improved the prediction performance when added to the existing prediction model.

**Strengths:**

* This paper deals with a practical problem of instantaneous trajectory prediction. The idea of using backwardly prediction and using it as complementary information proposed by the authors is novel. They also showed experimentally that as the amount of predicted complementary information increases, noise and redundancy increase, and it makes sense to adequately propose a module, NRRFormer to overcome this.

* The effectiveness of the proposed method was verified in two famous datasets and two prediction models. It also showed superior prediction performance compared to MOE and Distill, which dealt with the same topic.

* The paper is well-organized and easy to read. And the authors’ claim is somewhat well supported by experimental evidence.

**Weaknesses:**

Some details are missing.
* Why does $\hat{v}^{unobs}_1$ become mean of $V^{obs}$ on page 5? Is this mean for i=1,2 and all agents?
* There seems to be a lack of analysis on why cts loss enables better reconstruction on the last line of page 5. Personally, I think that if only recon loss is used, the network may fall into a trivial solution that creates the same unobserved representation regardless of time step and agent, and cts loss prevents this. I’m curious about the authors’ thoughts on this, and I think it would be good to add it to the manuscript.
* In the main result of Table 1, how was the baseline model (LaneGCN, HiVT) trained? The nuScenes and Argoverse prediction data already include data with short observation lengths. When training the baseline model, did you filter out data with full length for training, or did you filter out data with only 2 observations for use, or did you use all data?

**Questions:**

* In comparison experiments with MOE or Distill, they do not seem to use HiVT or LaneGCN as backbone. But isn’t MOE or Distill also plug-and-play? For example, Distill still seems to be able to applied on HiVT or LaneGCN while maintaining the encoder and decoder structure and doing knowledge distillation. It seems fair to compare with MOE or Distill using same backbones (HiVT and LaneGCN).
This may be the critical part for the fairness of the main experimental result, so if this is clarified, I think i can keep my rating more confidently.
* Trajectory prediction generally predicts multiple futures, not one future. The proposed backwardly prediction seems to predict only one past, but have you ever experimented when predicting multiple pasts?"

---

> ### Author Response · Authors · 2023-11-21
> **Response to Reviewer LWjU [1/1]**
>
> >Q1: Why does $\hat{v}^{unobs}_1$ become mean of $\mathbf{V}^{obs}$ on page 5? Is this mean for i=1,2 and all agents?
>
> R1: Sorry for confusing you. The Eq. (4) on Page 5 aims to predict unobserved trajectories $\hat{\mathbf{V}}^{unobs} = \\{\hat{v}\_0^{unobs}, \hat{v}\_{-1}^{unobs}, \cdots, \hat{v}\_{-N+1}^{unobs} \\}$ based on the observed trajectories $\mathbf{V}^{obs} = \\{v^{obs}\_1, v^{obs}\_2\\}$ through LSTM.  We use the symbol $\hat{v}_1^{unobs}=\textbf{Mean}(\mathbf{V}^{obs})$ as the initial input of LSTM. When $i$ is $0, -1, \cdots, -N+1$, we use Eq. (4) to calculate $\hat{v}^{unobs}\_{i}$. We will refine it in the final version.
>
> >Q2: There seems to be a lack of analysis on why cts loss enables better reconstruction on the last line of page 5. Personally, I think that if only recon loss is used, the network may fall into a trivial solution that creates the same unobserved representation regardless of time step and agent, and cts loss prevents this. I’m curious about the authors’ thoughts on this, and I think it would be good to add it to the manuscript.
>
> R2: Thanks for your comments. We would like to kindly clarify that if only the reconstruction loss is used, the network does not fall into a trivial solution, since we use the ground-truth unobserved features as supervised signals. The objective of the loss function $\mathcal{L}_{ctx}$ is to enhance the diversity of reconstructed features in unobserved historical trajectories. This, in turn, results in reduced redundancy among these features, thereby improving the accuracy in predicting future trajectories. We will refine it in the final version.
>
> > Q3: In the main result of Table 1, how was the baseline model (LaneGCN, HiVT) trained? The nuScenes and Argoverse prediction data already include data with short observation lengths. When training the baseline model, did you filter out data with full length for training, or did you filter out data with only 2 observations for use, or did you use all data?
>
> R3: Thanks for your comment. When training models,  we utilize the entire dataset, opting to truncate the observed trajectory points to 2 for the model input.
>
> > Q4: In comparison experiments with MOE or Distill, they do not seem to use HiVT or LaneGCN as backbone. But isn’t MOE or Distill also plug-and-play? For example, Distill still seems to be able to applied on HiVT or LaneGCN while maintaining the encoder and decoder structure and doing knowledge distillation. It seems fair to compare with MOE or Distill using same backbones (HiVT and LaneGCN). This may be the critical part for the fairness of the main experimental result, so if this is clarified, I think i can keep my rating more confidently.
>
> R4: Thanks for your comments. Based on Table 1, we observed that HiVT achieved better performance than LaneGCN on both Argoverse and nuScenes. Therefore, we used HiVT as the backbone of MOE and Distill for a fair comparison. The implementation details can be found in the Appendix A.6.
>
> Q5: Trajectory prediction generally predicts multiple futures, not one future. The proposed backwardly prediction seems to predict only one past, but have you ever experimented when predicting multiple pasts?
>
> R5: Thanks for your comment. In our work, we propose a feature reconstruction loss, in an effort to predict the features instead of waypoints for unobserved history trajectories. The reason is as follows: Our ITPNet first utilizes the Backward Forecast module to predict the features of unobserved history trajectories, and then combines them with those of observed trajectories for predicting future trajectories. Therefore,  we believe that achieving a high level of precision in predicting the features of unobserved historical trajectories will be more beneficial for future trajectory prediction than focusing on unobserved historical waypoints. Therefore, we opt to predict features rather than waypoints for the unobserved historical trajectories.
>
> When predicting the features of unobserved history trajectories, we simply choose the single-modal forecasting strategy. We think it is interesting to explore multi-modal method for predicting the features of unobserved history trajectories. This exploration may involve crafting appropriate loss functions tailored for multi-modal feature prediction.

---

> ### Comment · Reviewer_LWjU · 2023-11-22
>
> Thank you to the authors for addressing my questions.
>
> Regarding the employment of MOE and Distill backbones, I understand that HiVT is used as a backbone.
> However, considering the proposed method's plug-and-play nature, it would be valuable to also compare MOE and Distill with two backbone models and rearange the layout of Table 1 (LaneGCN + MOE / LaneGCN + Distill).
> By doing so, it would be more effective in showing the improvement of the proposed model compared to two previous works across various backbones.

---

> > ### Author Response · Authors · 2023-11-22
> > **Re：using LaneGCN as the backbone**
> >
> > Thanks for your advice. Following your suggestion, we performed another experiments by utilizing LaneGCN as the backbones of MOE and Distill. The results are presented in the table below. We can see that our proposed ITPNet still outperforms MOE and Distill, when using LaneGCN as the backbone. We will add them in the final version.
> > |         |           |      |  K=1 |     |      | K=6  |     |
> > |---------|---------|------|------|-----|------|------|-----|
> > |Dataset  | Method  |minADE|minFDE|minMR|minADE|minFDE|minMR|
> > |         |HiVT|4.158|8.368|0.846|1.085|1.712|0.249|
> > |         |LaneGCN|4.204|8.647|0.861|1.126|1.821|0.278|
> > |         |MOE+HiVT|3.312|6.840|0.794|0.939|1.413|0.177|
> > |         |Distill+HiVT|3.251|6.638|0.771|0.968|1.502|0.185|
> > | Argoverse|ITPNet+HiVT|2.631|5.703|0.757|0.819|1.218|0.141|
> > |         |MOE+LaneGCN|3.958|8.264|0.842|1.089|1.734|0.265|
> > | |Distill+LaneGCN|3.768|7.926|0.817|1.077|1.687|0.252|
> > | |ITPNet+LaneGCN|2.922|5.627|0.765|0.894|1.425|0.173|

---

### Official Review · Reviewer_1F4C · 2023-11-17

**Soundness:** 3 good
**Presentation:** 3 good
**Contribution:** 3 good
**Rating:** 6
**Confidence:** 3

**Summary:**

The authors address the task of trajectory prediction for autonomous driving when limited prior observations are given (such as when a newly tracked exo-vehicle appears from an obstruction). They note and experimentally show that existing methods – which typically assume a lengthy observations history (such as 2s or 20 discrete timesteps) - are ill-suited for this task (Figure 1). The authors show that this trend persists when the model is trained with few or many prior observation timesteps.

To ameliorate performance, they propose two adjustments (summarized in Figure 2) when forming the latent state to be used for downstream trajectory prediction:

1) They reconstruct previous timestep latent states corresponding to unobserved poses through a backward forecasting loss (section 3.3, equation 2). An additional loss term is also introduced encourage variability between latent states of different timesteps (equation 7).

2) They propose a self-attention module to limit redundancy and noise in the latent state representation dubbed NRRFormer (section 3.4)

The final latent state representation from their network is then input into a downstream trajectory prediction module (HiVT or LaneGCN).

Their approach is validated in trajectory prediction using the Argoverse and NuScene datasets using only 2 prior observations (section 4). They find improved ADE performance compared to baselines (section 4.4, Table 1). An ablation study is included in Table 2 assessing the effects of the proposed adjustments.

They further assess the effect of changing the number (N) of predicted unobserved prior poses (Table 3) where they find that performance gradually increases with N before dropping which they assert is caused by the introduction of noise and redundancy. Although they find that their NRRFormer potentially eliminates this issue.

Finally, they visually inspect the predicted trajectories (Figure 4) and find their approach to yield more diverse and accurate trajectories compared to baselines.

**Strengths:**

-	Tackles important research area (trajectory prediction under limit prior observations) often overlooked.
-	State-of-the-art performance.
-	Ablation study included.
-	Well written and easy to understand.

**Weaknesses:**

- The addition of the cts loss and NRRFormer in abalation study results (Table 2) appear to have very small / questionable performance gains. Given the small change, can the authors speak to the consistency of these results? Given the small change, were multiple network seeds or trials done and do the same improvements remain? I would have found it useful to report a confidence interval or variance over the results although perhaps it is not conventional in this area.

- It would have insightful to report how the NRRFormer affects performance for smaller values of N in Table 3. As it currently is, section 4.4 “Analysis of Different Lengths $N$” seems somewhat rushed with mentions of how the usage of the NRRFormer was done “without tuning it carefully”.

-	It would be useful to show how the method’s performance changes for different number of prior observed locations (T) since only 2 prior observations are considered in this work (Table 4). During practical usage, I would assume that we would want to use all available prior observed locations for future trajectory prediction and so the T value will change. The authors have shown that their method outperforms baselines at T=2 prior observations, but does this trend continue for higher values of T? Does the method improve performance at all values of T versus baselines or is there a point where it is a detriment. For example, given a test trajectory with T=10 prior observations, do we trust the author’s method over baselines?

- Although not needed at test time, the method requires ground-truth positions of unobserved states for the backward forecasting reconstruction loss during training. Depending on the dataset collection procedure, these may be hard to obtain. Furthermore – from what I understand – the HiVT and LaneGCN baselines in Table 1 are only trained with 2 observed prior locations and so it could be argued that the proposed approach requires more labeled data (predicts additional timesteps of prior locations which requires ground-truth labels). Although, at least for the HiVT method, the authors assert that training on all historic prior locations actually decreases performance (Figure 1) and so the second part of this criticism may be a moot. Nonetheless, I wonder if the additional labeled data could be used by the baselines in some other way (for example, training with variable length sequences).

- The approach assumes given 2d locations as prior observations instead of raw sensory input. For the problem cases that this work attempts to address (example: vehicle suddenly emerging behind obstruction), I would wonder how accurate these 2d locations may be given limited tracking timesteps. Noisy or inaccurate initial 2d poses may have negative downstream consequences when input into the authors’ method and thus reduce the reported performance gains that they assert in their results. From what I understand, the authors simply truncated longer fully observed trajectories to 2 observations and so the unique circumstances of the previous problem case may be ignored.

__Minor wording corrections to improve the final version (no effect on score):__

- On page 2, the usage of “straightly” in “Let’s consider a scenario where a vehicle travels straightly …” is awkward. Perhaps simply replace with the word “straight”.

**Questions:**

-	Did the authors try reconstructing raw 2d positions instead of their corresponding latent states (equation 2). Can they speak as to why one was done over the other?
-	The addition of the cts loss and NRRFormer in abalation study results (Table 2) appear to have very small / questionable performance gains. Given the small change, can the authors speak to the consistency of these results? Given the small change, were multiple network seeds or trials done and do the same improvements remain? I would have found it useful to report a confidence interval or variance over the results although perhaps it is not conventional in this area.
-	Is the margin parameter $\delta$ in equation 7 output by the network or a set hyper-parameter?
-	Can the authors clarify with how many prior observations the baselines were trained with in Table 1? Matching the results with Table 4 in the appendix, it appears to be 2, but I would appreciate if this was clarified.

---

> ### Author Response · Authors · 2023-11-21
> **Response to Reviewer 1F4C [1/3]**
>
> >Q1: The addition of the cts loss and NRRFormer in abalation study results (Table 2) appear to have very small / questionable performance gains. Given the small change, can the authors speak to the consistency of these results? Given the small change, were multiple network seeds or trials done and do the same improvements remain? I would have found it useful to report a confidence interval or variance over the results although perhaps it is not conventional in this area.
>
> R1: Thanks for your advice. Following your suggestion,  we further run the experiments 4 trials with different random seeds, and perform the t-test to verify that the improvements of the proposed components are statistically significant. The experimental results are listed in the table below. We can see that almost all p-values are less than the predetermined significance level of $\alpha=0.05$. This suggests that the performance is significantly improved, i.e., all the components in our method are effective for the instantaneous trajectory prediction task.
>
> For showcasing the effectiveness of our NRRFormer in more detail, please kindly refer to the response to Q3 for Reviewer ptje.
>
> |           |           |          |      |       | K=1  |       |     |       |
> |-          |-          |-         |-     |-      |-     |-      |-    | -     |
> | $L_{rec}$ | $L_{ctx}$ | NRRFormer|minADE|p-value|minFDE|p-value|minMR|p-value|
> |          |           |          |   $4.146±0.080$   |   -    |   $8.347±0.074$   |   -    |  $0.844±0.011$   |   -    |
> | √          |       |  |    $2.667±0.056$       |     $8.9\times10^{-6}$     |   $5.801±0.054$   |   $9.3\times10^{-7}$    |  $0.764±0.002$   |   $2.1\times10^{-4}$    |
> | √         | √         |          |  $2.615±0.013$    |    $0.049$   |   $5.737±0.016$   |    $0.030$   |  $0.761±0.001$   |   $0.029$    |
> | √         | √         |     √    |   $2.577±0.041$   |   $0.065$   |   $5.646±0.051$   |   $0.014$    |  $0.753±0.003$   |   $0.007$    |
>
>
>
> |           |           |          |      |       | K=6  |       |     |       |
> |-          |-          |-         |-     |-      |-     |-      |-    | -     |
> | $L_{rec}$ | $L_{ctx}$ | NRRFormer|minADE|p-value|minFDE|p-value|minMR|p-value|
> |          |           |          |   $1.089±0.003$   |   -    |   $1.719±0.006$   |   -    |  $0.248±0.002$   |   -    |
> | √          |       |  |    $0.843±0.003$       |     $2.4\times10^{-7}$     |   $1.293±0.007$   |   $5.0\times10^{-9}$    |  $0.155±0.002$   |   $1.9\times10^{-6}$    |
> | √         | √         |          |  $0.832±0.002$    |    $0.005$   |   $1.263±0.003$   |    $0.002$   |  $0.149±0.001$   |   $8.1\times10^{-4}$    |
> | √         | √         |     √    |   $0.822±0.002$   |   $9.4\times10^{-4}$   |   $1.222±0.003$   |   $2.0\times10^{-4}$    |  $0.143±0.001$   |   $3.9\times10^{-4}$    |
>
>
> >Q2: It would have insightful to report how the NRRFormer affects performance for smaller values of N in Table 3. As it currently is, section 4.4 "Analysis of Different Lengths" seems somewhat rushed with mentions of how the usage of the NRRFormer was done “without tuning it carefully”.
>
> R2: Thanks for your advice. Due to the deadline, we directly set $N=10$ as the length of predicted unobserved locations without tuning. Following your suggestion, we conducted experiments with more $N$. The experimental results are listed in the table below. We will add them in the final version.
>
> |  |           | | K=6| | | |K=6 | |
> |- |-          |-|-|-|-|-|-|-|
> | N| NRRFormer | minADE |minFDE | minMR| NRRFormer| minADE| minFDE| minMR|
> |0 |    ×      |1.068|1.678|0.241|-|-|-|-|
> |1 |    ×      |0.969|1.494|0.193|√|0.964|1.498|0.194|
> |2 |    ×      |0.872|1.329|0.160|√|0.868|1.323|0.158|
> |3 |    ×      |0.832|1.262|0.149|√|0.828|1.254|0.147|
> |4 |    ×      |0.845|1.291|0.154|√|0.824|1.240|0.146|
> |5 |    ×      |0.859|1.312|0.156|√|0.822|1.232|0.145|
> |6 |    ×      |0.867|1.302|0.161|√|0.823|1.231|0.145|
> |7 |    ×      |0.881|1.375|0.173|√|0.820|1.222|0.143|
> |8 |    ×      |0.903|1.410|0.181|√|0.821|1.222|0.142|
> |9 |    ×      |0.933|1.453|0.187|√|0.819|1.220|0.142|
> |10|    ×      |0.967|1.522|0.196|√|0.819|1.218|0.141|
>
> Based on the above table, we  have some interesting findings: When NRRFormer is not enabled, the prediction error initially decreases and then increases with the increase of $N$. This is attributed to the introduction of noise and redundancy when predicting a longer feature sequence. When NRRFormer is enabled, the performance of our method is consistently improved as $N$ increases. This illustrates our NRRFormer model indeed can filter out redundant and noisy information, demonstrating its effectiveness.

---

> ### Author Response · Authors · 2023-11-21
> **Response to Reviewer 1F4C [2/3]**
>
> >Q3: It would be useful to show how the method’s performance changes for different number of prior observed locations (T) since only 2 prior observations are considered in this work (Table 4). During practical usage, I would assume that we would want to use all available prior observed locations for future trajectory prediction and so the T value will change. The authors have shown that their method outperforms baselines at T=2 prior observations, but does this trend continue for higher values of T? Does the method improve performance at all values of T versus baselines or is there a point where it is a detriment. For example, given a test trajectory with T=10 prior observations, do we trust the author’s method over baselines?
>
> R3: Thanks for your comments. We would like to clarify that our ITPNet can take as input observed trajectories with varying lengths. This is achieved through the utilization of both LSTM in the backward forecasting module and the self-attention mechanism in the NRRFormer module. Following your suggestion, we added the experiments with different lengths of observed trajectories $T$, and compared our method with the two instantaneous trajectory prediction baselines, MOE and Distill. Our approach continues to demonstrate superior performance across various values of $T$ compared to the baselines.
>
> |         |  |         |      |  K=1 |     |      | K=6  |     |
> |---------|--|---------|------|------|-----|------|------|-----|
> |Dataset  |T | Method  |minADE|minFDE|minMR|minADE|minFDE|minMR|
> |         |  | MOE     |  3.312    |   6.840   |  0.794   |   0.939   |  1.413    |  0.177   |
> |         |2 | Distill |  3.251    |  6.638    |  0.771   |   0.968   |  1.502    |  0.185   |
> |         |  | ITPNet  |  2.631    |  5.703    |  0.757   |   0.819   |   1.218    |  0.141   |
> |         |  | MOE     |  2.562    |   5.607   |  0.776   |   0.784   |  1.221    |  0.134   |
> |Argoverse|5 | Distill |  2.465    |  5.452    |  0.756   |   0.796   |  1.248    |  0.139   |
> |         |  | ITPNet  |  2.410    |  5.257    |  0.738   |  0.748   |   1.132    |  0.122   |
> |         |  | MOE     |  2.357  |  5.141    |  0.733   |  0.726    |  1.101    |  0.117   |
> |         |10| Distill |   2.224    |  5.039    |  0.726   |  0.731    |  1.118    |  0.119   |
> |         |  | ITPNet  | 2.190   |   4.792   |  0.716   |  0.718    |  1.088    |  0.113   |
>
> >Q4: Although not needed at test time, the method requires ground-truth positions of unobserved states for the backward forecasting reconstruction loss during training. Depending on the dataset collection procedure, these may be hard to obtain. Furthermore – from what I understand – the HiVT and LaneGCN baselines in Table 1 are only trained with 2 observed prior locations and so it could be argued that the proposed approach requires more labeled data (predicts additional timesteps of prior locations which requires ground-truth labels). Although, at least for the HiVT method, the authors assert that training on all historic prior locations actually decreases performance (Figure 1) and so the second part of this criticism may be a moot. Nonetheless, I wonder if the additional labeled data could be used by the baselines in some other way (for example, training with variable length sequences).
>
> R4: Thanks for your comments. We agree that it might be expensive to obtain the ground-truth positions of agents. Nevertheless, please kindly note that while we necessitate ground-truth positions of unobserved trajectories during training, the overall count of ground-truth positions in our method is fewer than that of standard trajectory prediction methods (e.g., $T=20$ for Argoverse).
>
> Following your suggestion, we conducted another experiment to enhance all baselines by incorporating the unobserved trajectory locations as part of the supervised signal. Specifically, we integrated an additional decoder into the baseline to predict the locations of unobserved trajectories. The results presented in the table below continue to illustrate the superior performance of our method over all other instantaneous trajectory prediction methods.
>
> |         |         |      |  K=1 |     |      | K=6  |     |
> |---------|---------|------|------|-----|------|------|-----|
> |Dataset  | Method  |minADE|minFDE|minMR|minADE|minFDE|minMR|
> |         | MOE     |  3.248    |   6.715   |  0.783   |   0.927   |  1.391    |  0.174   |
> |Argoverse| Distill |  3.212    |  6.529    |  0.765  |   0.961   |  1.488    |  0.184   |
> |         | ITPNet+HiVT  |  2.631    |  5.703    |  0.757   |   0.819   |  1.218   |  0.141|

---

> ### Author Response · Authors · 2023-11-21
> **Response to Reviewer 1F4C [3/3]**
>
> >Q5: The approach assumes given 2d locations as prior observations instead of raw sensory input. For the problem cases that this work attempts to address (example: vehicle suddenly emerging behind obstruction), I would wonder how accurate these 2d locations may be given limited tracking timesteps. Noisy or inaccurate initial 2d poses may have negative downstream consequences when input into the authors’ method and thus reduce the reported performance gains that they assert in their results. From what I understand, the authors simply truncated longer fully observed trajectories to 2 observations and so the unique circumstances of the previous problem case may be ignored.
>
> R5: Thanks for your comments.  We performed another experiment where we added Gaussian noise $\mathcal{N}(0,\sigma)$ to two observed locations. The results provided in the table below demonstrate that our method still achieves superior performance compared to MOE and Distill, when the observed locations exhibit noise.
>
>
>
>
> |         |           |      |  K=1 |     |      | K=6  |     |
> |---------|---------|------|------|-----|------|------|-----|
> |Dataset  | Method  |minADE|minFDE|minMR|minADE|minFDE|minMR|
> |         |MOE                       |3.312|6.840   | 0.794  | 0.939 | 1.413  | 0.177  |
> |         |Distill                   |3.251|6.638   | 0.771  | 0.968 | 1.502  | 0.185  |
> |Argoverse|ITPNet+HiVT               |2.631|5.703   | 0.757  | 0.819 | 1.218  | 0.141  |
> |         |MOE($\sigma=0.1m$)        |3.426|7.114   | 0.836  | 1.002 | 1.549  | 0.202  |
> |         |Distill($\sigma=0.1m$)    |3.374|6.982   | 0.822  | 1.046 | 1.616  | 0.213  |
> |         |ITPNet+HiVT($\sigma=0.1m$)|2.938|6.424   | 0.792  | 0.909 | 1.347  | 0.165  |
>
> >Q6: On page 2, the usage of “straightly” in “Let’s consider a scenario where a vehicle travels straightly …” is awkward. Perhaps simply replace with the word “straight”.
>
> R6: Thanks for your advice.  We will revise it in the final version.
>
> >Q7: Did the authors try reconstructing raw 2d positions instead of their corresponding latent states (equation 2). Can they speak as to why one was done over the other?
>
> R7: Thanks for your comment. In our work, we predict the features instead of waypoints for unobserved history trajectories. The reason is as follows: Our ITPNet first utilizes the Backward Forecast module to predict the features of unobserved history trajectories, and then combine them with those of observed trajectories for predicting future trajectories.  Therefore,  we believe that achieving a high level of precision in predicting the features of unobserved historical trajectories will be more beneficial for future trajectory prediction than focusing on unobserved historical waypoints. Therefore, we opt to predict features rather than waypoints for the unobserved historical trajectories.
> Nevertheless, it is worthwhile to conduct further investigation into whether predicting features yields superior results compared to predicting waypoints, considering both methodological and empirical perspectives. Due to deadline, we will explore it in our future work.
>
> > Q8: Is the margin parameter in equation 7 output by the network or a set hyper-parameter?
>
> R8: Thanks for your comment. The margin is a hyper-parameter. We set it to $0.1$ throughout the experiments. We will clarify this in the final version.
>
> > Q9: Can the authors clarify with how many prior observations the baselines were trained with in Table 1? Matching the results with Table 4 in the appendix, it appears to be 2, but I would appreciate if this was clarified.
>
> R9: Thanks for your comment. Yes, the baselines were trained using two observations. We will clarify this in the final version.

---

> > ### Comment · Reviewer_1F4C · 2023-11-22
> >
> > Thank you for responding to my questions and addressing my concerns.

---

### Official Review · Reviewer_oKM3 · 2023-11-19

**Soundness:** 2 fair
**Presentation:** 3 good
**Contribution:** 2 fair
**Rating:** 5
**Confidence:** 5

**Summary:**

This work aims to solve the task where the observation is two points for motion prediction. They proposes to first reconstruct the unobserved longer history feature and then use them to update agent vector by attention. It could bring performance gains for existing works.

**Strengths:**

1.  According to the experiments, it indeed improves performance in this specific task.

2. It is plug-and-play for any trajectory prediction model, which could be useful.

**Weaknesses:**

1. The baselines is too old. Though LaneGCN and HiVT are both classic works, they are far from state-of-art-performace. Open sourced works like QCNet, MTR might worth trying.

2. Limited usage. The instantaneous trajectory prediction is interesting. However, the proposed method brings lots of extra parameters and computations.  Let's discuss an actual deployment problem: I observe that even ITPNet+HiVT < HiVT with 2s inputs and during your training, all parameters of HiVT are tuned without freezing. Thus, during actual deployment , the system should run an extra inference of the ITPNet+HiVT for those instantaneous objects while running the original HiVT for all the other fully observed agents. I am not sure whether worth it to double the inference for those instantaneous objects.

3. Some experiments and ablations are unclear, which seems that the work is incomplete and the working part is unclear. See question section.

**Questions:**

1. **One interesting perspective is that: the proposed method might benefit from the extra training signals of the task of predicting history instead of only future,  which could better utilize data like in Forcase-MAE [1].**  How would the authors think about it?

2. Why only single-mode forcasting for history instead of multi-mode like for future prediction?

3. Did you compare the results of predicting waypoints and predicting features?

4. The NRRFormer and backward forecasting  steps N=10 seems harmful for the best mode (K=1).

5. The contrastive loss seems have little influence. The authors might consider multiple tries.

[1] Forecast-MAE: Self-supervised Pre-training for Motion Forecasting with Masked Autoencoders. ICCV 23.

---

> ### Author Response · Authors · 2023-11-21
> **Response to Reviewer oKM3 [1/2]**
>
> >Q1: The baselines is too old. Though LaneGCN and HiVT are both classic works, they are far from state-of-art-performace. Open sourced works like QCNet, MTR might worth trying.
>
> R1: Thanks for your advice. In this work, we focus on studying the instantaneous trajectory prediction problem. Thus, we mainly compare ITPNet with the two most closely related approaches for instantaneous trajectory prediction, i.e., MOE and distill. LaneGCN and HiVT are two standard trajectory prediction models which need sufficient observed trajectories. Our method is plug-and-play and utilizes LaneGCN and HiVT as our backbones, respectively. We present the results of  LaneGCN and HiVT In Table 1, aiming to demonstrate that standard trajectory prediction methods struggle to effectively handle instantaneous trajectory prediction scenarios. Nevertheless, we believe it is a promising idea to integrate our method with state-of-the-art models, such as QCNet and MTR. In principle, our method can be compatible with QCNet and MTR. Due to the deadline, we plan to explore this in our future work.
>
> >Q2: Limited usage. The instantaneous trajectory prediction is interesting. However, the proposed method brings lots of extra parameters and computations. Let's discuss an actual deployment problem: I observe that even ITPNet+HiVT \textless HiVT with 2s inputs and during your training, all parameters of HiVT are tuned without freezing. Thus, during actual deployment , the system should run an extra inference of the ITPNet+HiVT for those instantaneous objects while running the original HiVT for all the other fully observed agents. I am not sure whether worth it to double the inference for those instantaneous objects.
>
> R2: Thanks for raising the concern which is helpful for clarifying the real-world deployment problem. We guess that this concern may arise from a misunderstanding of the inference process. We would like to clarify that our ITPNet performs the inference ONLY once, as our method can take as input observed trajectories with varying lengths. This is achieved through the utilization of both LSTM in the backward forecasting module and the self-attention mechanism in the NRRFormer module.
> Therefore, we have only one ITPNet+backbone model and conduct predictions with varying lengths of available observed trajectories in the real-world deployment, as mentioned by Reviewer ptje. Due to the deadline, we will perform such an experiment in the final version. Nevertheless, we believe that our method can achieve better performance in principle, as the involvement of more observed trajectories enables a more precise prediction of unobserved historical trajectories.
>
> Besides, please kindly note that the reason for fixing the number of observed points $T=2$ is that we follow the same problem setting as that of MOE [1], the first work in instantaneous trajectory prediction. MOE focuses on the most extreme scenario, where only 2 frames of locations can be observed.
> For a fair comparison, we also fix $T=2$.
>
> [1] Human Trajectory Prediction with Momentary Observation. CVPR'22
>
>
> > Q3: One interesting perspective is that: the proposed method might benefit from the extra training signals of the task of predicting history instead of only future, which could better utilize data like in Forcase-MAE [1]. How would the authors think about it?
>
> R3: Thanks for your advice. Forecast-MAE is a self-supervised pre-training method designed for trajectory prediction, on the basis of the mask autoencoder framework. We believe it is a good idea to apply Forecast-MAE to our method. We can utilize Forecast-MAE to pre-train the encoder and the decoder in our method. Subsequently, we can fine-tune our method for the instantaneous trajectory prediction task. It is worth exploring it in our future work.

---

> ### Author Response · Authors · 2023-11-21
> **Response to Reviewer oKM3 [2/2]**
>
> >Q4: Why only single-mode forecasting for history instead of multi-mode like for future prediction?
>
> R4: Thanks for your comment. In our work, we propose a feature reconstruction loss, in an effort to predict the features instead of waypoints for unobserved history trajectories. The reason is as follows: Our ITPNet first utilizes the Backward Forecast module to predict the features of unobserved history trajectories, and then combines them with those of observed trajectories for predicting future trajectories.  Therefore,  we believe that achieving a high level of precision in predicting the features of unobserved historical trajectories will be more beneficial for future trajectory prediction than focusing on unobserved historical waypoints. Therefore, we opt to predict features rather than waypoints for the unobserved historical trajectories.
>
> When predicting the features of unobserved history trajectories, we simply choose the single-modal forecasting strategy. We think it is interesting to explore multi-modal method for predicting the features of unobserved history trajectories. This exploration may involve crafting appropriate loss functions tailored for multi-modal feature prediction.
>
> > Q5: Did you compare the results of predicting waypoints and predicting features?
>
> R5: Thanks for your comment. In our work, we predict the features instead of waypoints for unobserved history trajectories. The reason is as follows: Our ITPNet first utilizes the Backward Forecast module to predict the features of unobserved history trajectories, and then combines them with those of observed trajectories for predicting future trajectories. Therefore,  we believe that achieving a high level of precision in predicting the features of unobserved historical trajectories will be more beneficial for future trajectory prediction than focusing on unobserved historical waypoints. Therefore, we opt to predict features rather than waypoints for the unobserved historical trajectories.
> Nevertheless, it is worthwhile to conduct further investigation into whether predicting features yields superior results compared to predicting waypoints, considering both methodological and empirical perspectives. Due to deadline, we will explore it in our future work.
>
> >Q6: The contrastive loss seems have little influence. The authors might consider multiple tries.
> The NRRFormer and backward forecasting steps N=10 seems harmful for the best mode (K=1).
>
> R6: Thanks for your advice. Following your suggestion,  we further run the experiments 4 trials with different random seeds, and perform the t-test to verify that the improvements of the proposed components are statistically significant. The experimental results are listed in the table below. We can see that almost all p-values are less than the predetermined significance level of $\alpha=0.05$. This suggests that the performance is significantly improved, i.e., all the components in our method are effective for the instantaneous trajectory prediction task.
>
> For showcasing the effectiveness of our NRRFormer in more detail, please kindly refer to the response to Q3 for Reviewer ptje.
>
> |           |           |          |      |       | K=1  |       |     |       |
> |-          |-          |-         |-     |-      |-     |-      |-    | -     |
> | $L_{rec}$ | $L_{ctx}$ | NRRFormer|minADE|p-value|minFDE|p-value|minMR|p-value|
> |          |           |          |   $4.146±0.080$   |   -    |   $8.347±0.074$   |   -    |  $0.844±0.011$   |   -    |
> | √          |       |  |    $2.667±0.056$       |     $8.9\times10^{-6}$     |   $5.801±0.054$   |   $9.3\times10^{-7}$    |  $0.764±0.002$   |   $2.1\times10^{-4}$    |
> | √         | √         |          |  $2.615±0.013$    |    $0.049$   |   $5.737±0.016$   |    $0.030$   |  $0.761±0.001$   |   $0.029$    |
> | √         | √         |     √    |   $2.577±0.041$   |   $0.065$   |   $5.646±0.051$   |   $0.014$    |  $0.753±0.003$   |   $0.007$    |
>
>
>
> |           |           |          |      |       | K=6  |       |     |       |
> |-          |-          |-         |-     |-      |-     |-      |-    | -     |
> | $L_{rec}$ | $L_{ctx}$ | NRRFormer|minADE|p-value|minFDE|p-value|minMR|p-value|
> |          |           |          |   $1.089±0.003$   |   -    |   $1.719±0.006$   |   -    |  $0.248±0.002$   |   -    |
> | √          |       |  |    $0.843±0.003$       |     $2.4\times10^{-7}$     |   $1.293±0.007$   |   $5.0\times10^{-9}$    |  $0.155±0.002$   |   $1.9\times10^{-6}$    |
> | √         | √         |          |  $0.832±0.002$    |    $0.005$   |   $1.263±0.003$   |    $0.002$   |  $0.149±0.001$   |   $8.1\times10^{-4}$    |
> | √         | √         |     √    |   $0.822±0.002$   |   $9.4\times10^{-4}$   |   $1.222±0.003$   |   $2.0\times10^{-4}$    |  $0.143±0.001$   |   $3.9\times10^{-4}$    |

---

### Meta-Review · Area_Chair_wqWe · 2023-12-09

**Metareview:**

This paper proposes a trajectory prediction method for self-driving cars that have access to an HD map and have a very short history of observations (two timesteps) from which they have to make predictions. There are only a handful of papers in this direction, so it is an emerging task in computer vision for self-driving, although it also seems like a narrow problem definition. The paper advocates for predicting the past history of the trajectory in latent space from the two-frame observations and feeding the whole trajectory into an existing prediction backbone. The paper is well-written and experimental results support the claims.

I am not inclined to recommend this paper for acceptance because it addresses a narrow problem definition with more-or-less standard methodology in computer vision, and while the paper is well-executed, I don't think it has sufficient novelty in terms of backward infilling and forward prediction. In addition, the authors mention in their responses here that "the reason for fixing the number of observed points T = 2 is that we follow the same problem setting as that of MOE [1], the first work in instantaneous trajectory prediction." Even though a previous paper followed this evaluation protocol, it would make sense to allow for more than two observations, for the model to be more general as an infilling + prediction model. This, however, will require one more iteration of improvements over the current paper.

**Justification For Why Not Higher Score:**

See rationale above.

**Justification For Why Not Lower Score:**

N/A

---

### Decision · Program_Chairs · 2024-01-16

Reject